# Quantitative Evaluation of Inflammatory Markers in Peri-Implantitis and Periodontitis Tissues: Digital vs. Manual Analysis—A Proof of Concept Study

**DOI:** 10.3390/medicina58070867

**Published:** 2022-06-29

**Authors:** Dolaji Henin, Luiz Guilherme Fiorin, Daniela Carmagnola, Gaia Pellegrini, Marilisa Toma, Aurora Cristofalo, Claudia Dellavia

**Affiliations:** 1Department of Biomedical, Surgical and Dental Sciences, University of Milan, 20133 Milan, Italy; dolaji.henin@unimi.it (D.H.); luiz.fiorin@unimi.it (L.G.F.); gaia.pellegrini@unimi.it (G.P.); marilisa.toma@unimi.it (M.T.); auro.1995@hotmail.it (A.C.); claudia.dellavia@unimi.it (C.D.); 2Department of Diagnosis and Surgery, Division of Periodontics, School of Dentistry, Sao Paulo State University (UNESP), Aracatuba 16015-050, SP, Brazil

**Keywords:** peri-implantitis, digital pathology, histomorphometry, immunohistochemistry

## Abstract

*Background and Objectives*: In dentistry, the assessment of the histomorphometric features of periodontal (PD) and peri-implant (PI) lesions is important to evaluate their underlying pathogenic mechanism. The present study aimed to compare manual and digital methods of analysis in the evaluation of the inflammatory biomarkers in PI and PD lesions. *Materials and Methods*: PD and PI inflamed soft tissues were excised and processed for histological and immunohistochemical analyses for CD3+, CD4+, CD8+, CD15+, CD20+, CD68+, and CD138+. The obtained slides were acquired using a digital scanner. For each marker, 4 pictures per sample were extracted and the area fraction of the stained tissue was computed both manually using a 594-point counting grid (MC) and digitally using a dedicated image analysis software (DC). To assess the concordance between MC and DC, two blinded observers analysed a total of 200 pictures either with good quality of staining or with non-specific background noise. The inter and intraobserver concordance was evaluated using the intraclass coefficient and the agreement between MC and DC was assessed using the Bland–Altman plot. The time spent analysing each picture using the two methodologies by both observers was recorded. Further, the amount of each marker was compared between PI and PD with both methodologies. *Results:* The inter- and intraobserver concordance was excellent, except for images with background noise analysed using DC. MC and DC showed a satisfying concordance. DC was performed in half the time compared to MC. The morphological analysis showed a larger inflammatory infiltrate in PI than PD lesions. The comparison between PI and PD showed differences for CD68+ and CD138+ expression. *Conclusions:* DC could be used as a reliable and time-saving procedure for the immunohistochemical analysis of PD and PI soft tissues. When non-specific background noise is present, the experience of the pathologist may be still required.

## 1. Introduction

Dental implants are commonly used to replace missing teeth, as they provide safe and predictable outcomes in oral rehabilitation procedures [1]. Their long-term success is due to the correct integration of the implant surface with both hard and soft tissues, creating a biological seal that allows proper osseointegration and prevents bacterial invasion. The lacking or rupture of this barrier could lead to peri-implantitis (PI), an inflammatory disease that could evolve in the loss of the peri-implant-supporting bone [2].

PI shares some common features with periodontal disease (PD), an inflammatory plaque-induced condition affecting the supporting tissues of the teeth, which is host-mediated and might progressively lead to tooth loss if left untreated [3,4]. Compared to PD, PI generally shows a specific circumferential pattern of bone loss and altogether a faster progressive trend towards implant loss [2]. Previous reports have investigated the histopathological features of PI alone [5] or compared to PD. From a histopathological point of view, PI and PD are peculiar lesions due to their anatomical localisation adjacent to the junctional epithelium and to their composition [6]. They are similar in cellular composition, except for the predominance of plasma cells and lymphocytes and the larger portion of polymorphonuclear leukocytes and macrophages in PI [2]. Additionally, PI presents a considerably larger infiltrate than PD in the connective tissue [6]. Despite recent improvements in the immunopathological assessments of peri-implant soft tissue lesions [7], the pathogenetic mechanism of peri-implant disease has not yet been fully understood. From a clinical point of view, PI’s management is challenging because it has a fast and not linear progression, and it may involve an unpredictable individual predisposition [8].

Originally, like in other fields, the histopathological analysis and quantification of the oral tissues supporting the teeth and implants were performed by observing the sample images using a microscope and manually counting (MC) the target components by means of a superimposed grid, as described by Berglundh (1991) [9,10]. The grid could be adapted according to the researchers’ needs, by increasing or decreasing the number of intersection points. With time, digital enhancements have been introduced to facilitate and improve the image acquisition procedures, for example using high-resolution slide scanners, leading to digital pathology (DP) [11]. High-resolution scanners provide images that can be consulted immediately on the computer or stored in digital archives, always maintaining the same quality and not degrading over time. The images can be improved according to the researchers’ needs and browsed and expanded at high magnifications such as in optical microscopy. Thanks to these advantages, the use of digital scanners for slide acquisition has become common in pathology wards and in research, as it represents a time-saving and user-friendly procedure [12,13].

Further developments of computer-aided technologies and specific image analysis programs have led to new options and possibilities in order to ease histomorphometrical procedures, for example by introducing digital counting (DC) [14,15,16].

MC is a reliable procedure, although factors such as the experience of the observer can alter the number of measured cells, even after years of training [17,18,19]. Further, the distinction between the normal and hypocellularity conditions by different pathologists was associated with a low reproducibility rate [20,21].

The DC method relies on the use of a dedicated image analysis software and algorithms for the identification and measurement of the target tissue area components, and in some fields, such as the recognition of oncologic tissues, it is nowadays successfully used, as it provides accuracy, reproducibility, and standardisation in the analysis [12,13,22,23].

The use of DC on digitally scanned whole slide images has also been reported in dentistry [24], but its reliability for the analysis of peri-implant and periodontal lesions has not been assessed yet.

The aim of the present study was to evaluate the agreement between DC and MC for the immunohistochemical analysis of PI and PD soft tissues.

## 2. Materials and Methods

The patients were enrolled according to the following inclusion criteria:-Over 18 years old;-No systemic diseases;-No long-term medications;-No smoking;-No pregnant or lactating women;-Diagnosis of active periodontitis or peri-implantitis according to the criteria further described and requiring surgical therapy;-Have read and signed the informed consent.

Each patient provided one PI or PD site, with the following inclusion criteria:
-Hyperemia;-Bleeding;-Probing depth ≥ 5 mm;-Suprabony defect confirmed by intraoral radiography.

### 2.1. Surgical Procedures and Sample Collection

At surgery, after local anaesthesia was obtained, a full-thickness periodontal flap using a modified Widman technique was performed in PD patients and the excess tissue around the tooth was collected [25]. In PI patients, a full-thickness periodontal flap was performed and the inflammatory tissue surrounding the implants was excised and collected.

### 2.2. Histological and Immunohistochemical Processing

Immediately after surgery, the 2 × 3 mm biopsied specimens were immersion-fixed in 4% formalin/0.1 M phosphate-buffered saline (PBS) (pH 7.4) for 24 h at room temperature, then routinely dehydrated in increasing concentrations of ethanol (from 50 to 100%) and xylol and finally embedded in paraffin. Serial 4–5 µm buccal–lingual sections were obtained, mounted on 3-amino-propyl-trietoxi-xilane-coated slides, then hydrated in decreasing concentrations of xylol and ethanol (from 100 to 70%) and finally immersed in distilled water. The sections were stained with haematoxylin and eosin (HE) to evaluate the tissue morphology. For the immunohistochemical evaluation, the following markers were chosen according to previous studies [10,24] (Table 1): CD3+ (F7.2.38, NeoMarkers, ThermoFisher Scientific, Monza, Italy) for T lymphocytes, CD4+ (1F6, NovoCastra, Leica Biosystem, Milano, Italy) for T helper lymphocytes, CD8+ (C8/144B, DAKO, Agilent Technologies Italia S.p.A., Cernusco sul Naviglio, Italy) for T cytotoxic lymphocytes, CD15+ (C3D-1, DAKO, Agilent Technologies Italia S.p.A., Cernusco sul Naviglio, Italy) for neutrophils, CD20+ (L26, DAKO, Agilent Technologies Italia S.p.A., Cernusco sul Naviglio, Italy) for B lymphocytes, CD68+ (PGM1, DAKO, Agilent Technologies Italia S.p.A., Cernusco sul Naviglio, Italy) for macrophages, and CD138+ (B-B4, DBA Italia, Segrate, Italy) for plasma cells.

The sections were counterstained with Mayer’s haematoxylin, dehydrated, cover-slipped, and captured using a high-resolution digital scanner (NanoZoomer S60 ©, Hamamatsu Photonics KK, Hamamatsu City, Japan). Whole slide images were browsed using NanoZoomerDigitalPathology © software NDP.view v2.7 (Hamamatsu Photonics KK, Hamamatsu City, Japan) from 50× to 400× magnification [24,26].

The images were displayed on a monitor (LG PC IPS 24” 16:9 Full HD 1920 × 1080), calibrated using an Eye-One calibration kit (X-Rite, Tewksbury, Massachusetts, USA), and analysed. The morphometric evaluation, using both MC and DC, was performed twice (T1, T2) by two blinded observers (DH, LF), using the same computer and the same ambient light in order to avoid any influence on the overall assessment [27].

### 2.3. Selection and Analysis of the Images

The HE-stained slides were first examined to analyse the tissue structure and to define the localisation of the inflammatory lesion. From every immune-stained slide, 4 different pictures from the inflammatory connective tissue adjacent to the epithelial junction area were chosen and magnified at 400×.

#### 2.3.1. MC

The MC method consisted of manually counting the stained cells. A 594-dot grid was chosen in order to exploit a high number of intersection points. The grid was superimposed on each image and analysed using ImageJ (Rasband, W.S., ImageJ, U.S. National Institutes of Health, Bethesda, MD, USA). The intersection dots that fell on brown-coloured cytoplasm or membrane were counted. The counted dots were expressed as an estimated percentage of the area [28].

#### 2.3.2. DC

The DC required pixel standardisation before proceeding with the automatised count [29]. An image analysis protocol was, therefore, developed using Adobe Photoshop 6 © (Knoll, Adobe Inc., California, U.S). The protocol consisted of selecting a marked cell, magnifying the image, and selecting the colour of the stained cytoplasm or membrane with the colour range tool (select > colour range). This tool allows the selection of the pixels of a specific colour or a colour range in the whole image. The dark-brown cytoplasm and the nucleolus or membrane were selected with the eyedropper tool (select: sampled colours, fuzziness 100). This criterion of selection was saved and later applied to the other slides. The ratio of the selected pixels/total pixels was assessed utilizing the histogram tool (window > histogram) and the result was expressed as a percentage of the total pixels of the image.

#### 2.3.3. Agreement between MC and DC

To assess the agreement between the two methods, the two operators measured 100 random pictures (400× magnification) in which the expression of the markers was well-defined and characterised by neglectable background noise (GOOD samples group) and 100 random pictures in which the immunohistochemistry slides presented non-specific background noise (NOISE samples group) (Figure 1).

Further, the two observers analysed all PI and PD sections twice using both MC and DC.

The time spent analysing each picture using the two methodologies by both observers was recorded.

### 2.4. Statistical Analysis

#### 2.4.1. Comparison between Methods (MC vs. DC)

The intraobserver (T1 vs. T2) and interobserver (DH vs. LF) agreements were analysed using the intraclass correlation coefficient (ICC). GOOD and NOISE slides were counted in the same order and paired. Following the classification used by Fleiss (1986) [30], the reproducibility of the methodology was scored, varying between bad and excellent. The agreement between the two methods was assessed using a Bland–Altman plot with the upper and lower 95% confidence interval limits for the average difference, as well as the degree of concordance between measures. The correlation of the methods was performed using the Pearson correlation coefficient in order to define the statistical causal or non-causal relationship between the samples.

#### 2.4.2. Lesion Comparison (PI vs. PD)

The D’Agostino-Pearson test was performed to evaluate the normality of the data. For each variable, the differences between PD and PI were analysed using a Mann–Whitney test for non-parametric data. Finally, an intergroup analysis (MC vs. DC) for each marker was performed using the signed-rank test. For all tests, the level of significance was set at *p* < 0.05. 

## 3. Results

Twenty-two patients were enrolled, 11 with PI and 11 with PD, who provided a total of 22 samples, as each patient provided one biopsy.

### 3.1. Methodological Comparisons

#### 3.1.1. Concordance between Observers

The intraobserver reproducibility was excellent, and the interobserver reproducibility varied between good and excellent (ICC values are displayed in Table 2 and Table 3). While the intraobserver evaluation provided an excellent correlation for both DC and MC when considering all sections, the interobserver correlation provided excellent concordance in all but “noise” sections with DC. Nevertheless, the DC “noise” group provided a good correlation.

#### 3.1.2. Agreement and Correlation between Methods

##### Agreement

In all cases the average difference between MC and DC was offset from zero, meaning that there was a mean bias. Most of the points were within the acceptable limits of the confidence interval, suggesting that there was an agreement between both techniques despite the condition (GOOD or NOISE) of the analysed slides. In the GOOD slides group, 95% of the points were inside the confidence interval, presenting an average bias of −0.44% (Figure 2A). In the NOISE slides group, 92% of the points were inside the confidence interval, presenting an average bias of 0.25% (Figure 2B). In both plots, the data were clustered on the left and the points spread as they moved to the right, demonstrating a trend where higher values presented a higher mean bias [31].

##### Correlation

Both methodologies revealed a low coefficient of correlation (−0.08 for the GOOD group and −0.01 for the NOISE one), showing no linear correlation between the samples of the same slide group.

#### 3.1.3. Time

The average time spent by DH to analyse the images using DC was 0.51 min, compared to 2.40 min using MC. LF spent an average time of 1.05 min using DC and 2.03 min using MC.

### 3.2. Lesion Comparison

#### 3.2.1. Tissue Structure

##### PD Lesions

At low magnification, a mild alteration of the architecture of the tissue was observed, as some samples showing the presence of an ulcerated epithelium and hosting a dense inflammatory infiltrate (Figure 3A). At higher magnification, lymphocytes, polymorphonuclear cells, and plasma cells were distinguishable in the connective tissue. The vessels’ diameter seemed to be increased and displayed a reactive endothelium. No signs of necrosis or fibrosis were appreciable (Figure 3C).

##### PI Lesions

At low magnification, the PI samples seemed very similar to the PD ones. The peri-implant soft tissues showed some architectural alterations; the epithelial papillae presented an altered morphology and displayed a cellular infiltrate in the epithelium (Figure 3B), while a dense inflammatory infiltrate was observed in the connective tissue. The inflammatory infiltrate was considerably larger than the one observed in the PD lesions and was mainly characterised by plasma cells, lymphocytes, and polymorphonuclear cells. The PI tissue seemed more sclerotic than the PD one, with irregularly organised collagen fibers and less fibroblasts and vessels (Figure 3D).

#### 3.2.2. Immunohistochemical Analysis

The immunohistochemical analysis highlighted the different cellular composition of the inflammatory infiltrate. Figure 4 and Figure 5 show that CD4+, CD8+, CD15+, CD68+, and CD138+ were expressed on the cell membrane, while CD3+ and CD20+ were expressed both on the cell membrane and in the cytoplasm.

#### 3.2.3. Morphometry

##### MC

Figure 6 shows the morphometric analysis reporting the results of the different markers, expressed as percentages of the tissue area in PI and PD obtained by MC. The PD and PI lesions showed statistically significant differences for CD68+ and CD138+ staining. For CD3+, CD4+, CD8+, CD15+, and CD20+, no statistical differences were found.

##### DC

Figure 7 shows the morphometric analysis reporting the results of the different markers, expressed as percentages of the tissue area in PI and PD obtained by DC. The PD and PI lesions showed statistically significant differences for CD68+ and CD138+ staining. For CD3+, CD4+, CD8+, CD15+, and CD20+, no statistical differences were found.

##### MC vs. DC

An intergroup analysis between MC and DC for each marker was performed through the signed-rank test and resulted in no statistical differences.

## 4. Discussion

The present study aimed to compare the use of MC and DC in a very challenging dental issue, namely peri-implantitis, evaluating the inflammatory pattern by means of immunohistochemistry.

DC presented excellent intraobserver reproducibility and a strong agreement with MC, suggesting that within the context of the measurements performed in this study, DC might be an efficient and time-saving method for histomorphometric evaluations. DC is being implemented in the medical field, as some studies have shown that it presents several advantages, such as the simplicity and repeatability of the process, good ergonomics, immediate access to slides, possibility of remote sharing of information, and good accuracy concerning the results. DC was performed in the current study by means of a widely spread image analysis software, Adobe Photoshop © [23,32,33], which thanks to its versatility and availability has been adapted for use in the medical, dental, and pathological fields. This program allows colour analysis through the measurement of the number of pixels of a chosen colour range. However, the software might provide unreliable data when applied to images with background noise, most likely due to non-specific binding or in the presence of antibodies in different cellular localisations. In such situations, conversely an operator is able to discriminate cells from background noise [34,35].

In the present study, some slides presented a background noise (NOISE group) while others did not (GOOD group). Both MC and DC showed excellent intra- and interobserver concordance except for the NOISE group analysed by DC, which presented a good interobserver concordance. These results confirm that despite DC providing accurate results, a slide analysis should always include the supervision of a pathologist. In this scenario, the role of the pathologist could involve an initial evaluation and the selection of the slides according to their staining quality, the choice of the most appropriate method to analyse the images, and the final quality control of the procedure.

The results from the present study indicate that DC required half the time of MC, as described by other authors [36,37]. Time-saving can be a critical issue when a large number of samples has to be analysed, not only because long observation and analysis times can be rather expensive for the laboratory, but they can also be exhausting for the pathologist, potentially leading to errors. DC, therefore, can play a fundamental role in speeding up the process of acquisition and analysis.

MC and DC also provided comparable also concerning the evaluation of the inflammatory patterns of PD and PI. The inflammatory markers of PD and PI soft tissues used in this study have been investigated by other researchers, and their profiles could, therefore, be compared. Our results concerning the inflammatory cell measurements of PD and PI lesions obtained by both DC and MC are similar to those previously reported in the literature [6]. A significant difference in CD68+ and CD138+ expression between PI and PD was evidenced, while in agreement with Gualini and Berglund (2003) [38], no difference was evidenced regarding CD3+, CD4+, CD8+, and CD20+ expression levels. Concerning CD15+ expression, no statistical difference was evidenced between PI and PD lesions in the present study. De Araujo et al. (2017) [39] reported a higher expression level of CD15+ in PI lesions compared to healthy peri-implant tissues. In parallel, Dutzan et al. (2016) [40] reported increased levels of CD15+ in the crevicular fluid in periodontitis compared to healthy periodontal tissues.

The morphological observation of the samples included in the present study is in agreement with results previously reported in the literature. The architecture of the tissues showed limited differences between the two lesions, as both presented severe signs of inflammation [6], with PI tissues showing a larger inflammatory infiltrate lateral to the epithelial junction than PD lesions.

In the present study, some limitations have to be acknowledged. First, we applied DC and MC methods only on scanned images. Further investigations are needed to compare DC and MC in images obtained by optical microscope, since digital scanners are not available in all laboratories. Moreover, our results are based on 11 patients per group, prompting a call for larger studies in order to confirm our data. Additionally, DC could be applied to evaluate different colour ranges or tissue fractions, such as collagen and vessel contents by means of histochemical staining, opening broad application possibilities for other types of evaluation, such as evaluations of the quality of osseointegration, bone remodelling, and the outcomes of tissue engineering [10,41].

## 5. Conclusions

In conclusion, both the digital and manual analyses of scanned slides of PI and PD lesions provided similar results in terms of marker counting, although MC performed better than DC in images showing some background noise. DC on the other hand is time-efficient, and if correctly applied could turn out to be a valid and economic choice.

## Figures and Tables

**Figure 1 medicina-58-00867-f001:**
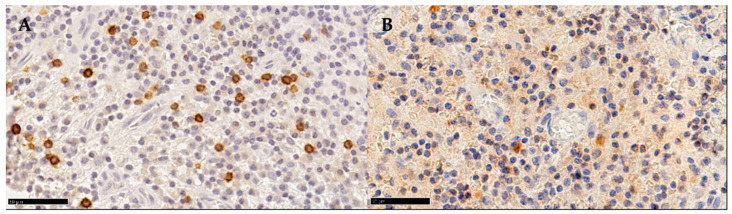
Detail of two CD4+ immunostained slides of PD. Exemplification of a (**A**) GOOD sample and (**B**) NOISE sample.

**Figure 2 medicina-58-00867-f002:**
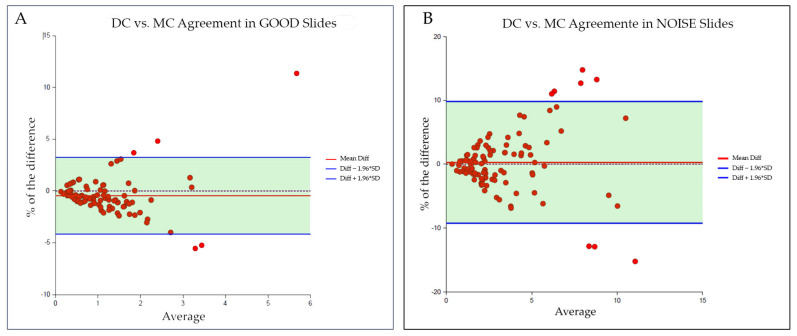
Bland–Altman plot of the paired samples, bias, and limits of agreement: (**A**) agreement between DC and MC within GOOD slides; (**B**) agreement between DC and MC within NOISE slides.

**Figure 3 medicina-58-00867-f003:**
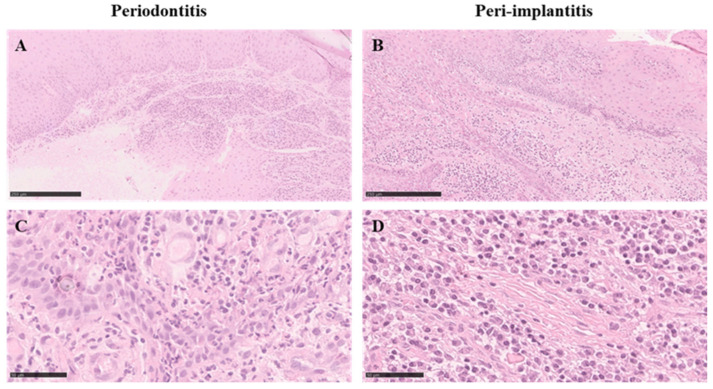
Sections of periodontitis and peri-implantitis tissues stained with HE: (**A**,**B**) low magnification; (**C**,**D**) high magnification.

**Figure 4 medicina-58-00867-f004:**
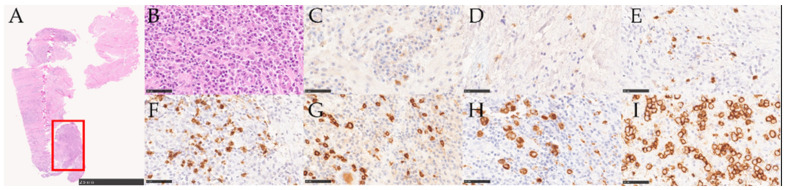
Stained samples of a PI lesion: (**A**) overview of a section stained with HE; (**B**) detailed view of (**A**); (**C**) CD3+; (**D**) CD4+; (**E**) CD8+; (**F**) CD15+; (**G**) CD20+; (**H**) CD68+; (**I**) CD138+.

**Figure 5 medicina-58-00867-f005:**
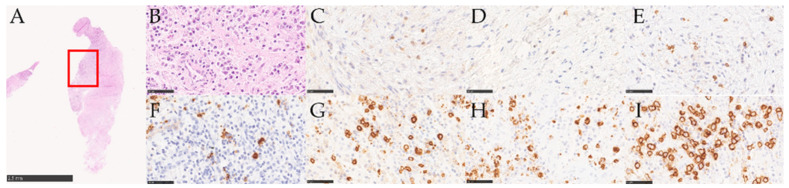
Stained samples of a PD lesion: (**A**) overview of a section stained with HE; (**B**) detailed view of (**A**); (**C**) CD3+; (**D**) CD4+; (**E**) I CD8+; (**F**) CD15+; (**G**) CD20+; (**H**) CD68+; (**I**) CD138+.

**Figure 6 medicina-58-00867-f006:**
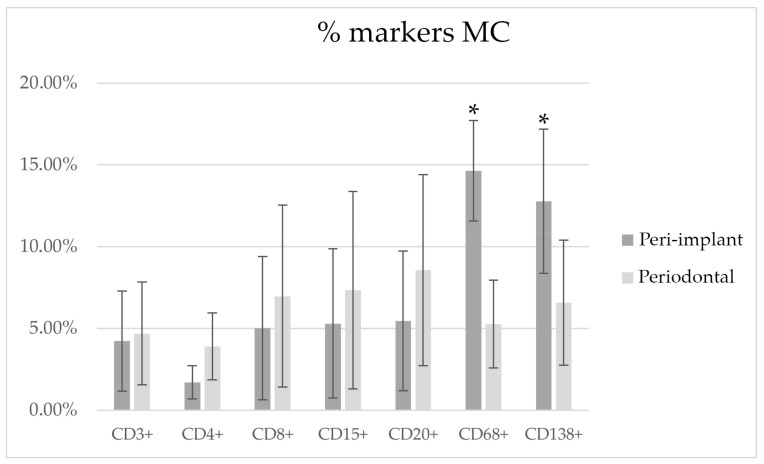
Markers counted with the MC technique. * Statistical difference evidenced between groups (*p* < 0.05).

**Figure 7 medicina-58-00867-f007:**
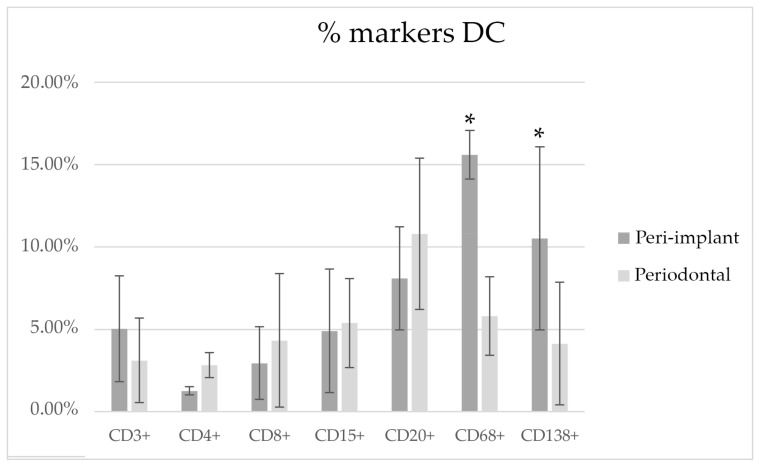
Markers counted with the DC technique. * Statistical difference evidenced between groups (*p* < 0.05).

**Table 1 medicina-58-00867-t001:** Characteristics of the antibodies and relative cellular types chosen for immunohistochemistry.

Cellular Type	Antibody	Clone	Diluition IgG	Pre-Treatment
T lymphocytes	CD3+	F7.2.38	1:40	EDTA
T helper	CD4+	1F6	1:50	EDTA
T cytotoxic	CD8+	C8/144B	1:100	Citrate
Neutrophils	CD15+	C3D-1	1:50	EDTA
B lymphocytes	CD20+	L26	1:500	Citrate
Macrophages	CD68+	PGM1	1:100	Citrate
Plasma cells	CD128+	B-B4	1:1000	Citrate

**Table 2 medicina-58-00867-t002:** Intraobserver ICC.

Observer	Technique	Variance	Error	*p*-Value	ICC	Fleiss (1986) [30]
Observer 1	DC	2.12	0.24	*p* < 0.0001	0.88	Excellent
	MC	1.52	0.10	*p* < 0.0001	0.77	Excellent
Observer 2	DC	2.28	0.37	*p* < 0.0001	0.85	Excellent
	MC	2.71	0.20	*p* < 0.0001	0.88	Excellent

**Table 3 medicina-58-00867-t003:** Interobserver ICC.

Technique	Background	Variance	Error	*p*-Value	ICC	Fleiss (1986) [30]
**DC**	GOOD	2.0	0.25	*p* < 0.0001	0.78	Excellent
	NOISE	2.01	0.41	*p* < 0.0001	0.65	Good
**MC**	GOOD	1.07	0.14	*p* < 0.0001	0.76	Excellent
	NOISE	1.82	0.12	*p* < 0.0001	0.87	Excellent

## Data Availability

The data that support the findings of this study are available from the corresponding author upon reasonable request.

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
