# Peer review of "Quantitative Evaluation of Inflammatory Markers in Peri-Implantitis and Periodontitis Tissues: Digital vs. Manual Analysis—A Proof of Concept Study"

_medicina, 2022, doi:10.3390/medicina58070867_

Round 1

Reviewer 1 Report

The submitted manuscript is aimed to compare two different methods for oral tissue evaluation, in particular for periodontitis and periimplantitis.

The study is well organized and well conducted, and of wide interst for both dentists and pathologists.

However, the authors enrolled 22 patients only, so it is a pilot study (or a proof of concept ctudy). This should be clear for the readers, and possibly reported in the title.

No specific ethical issues are present in the reported study, but  please specify the University name and the date of the protocol approval at l. 98

In sect. 2.2: please better specify the data regarding the manufacturers

In sect. 3.1.1: comment more extensively the results and the tables' content.

In Discussion, the authors must report the limits of the study.

At last, a revision for English language is suggested.

Author Response

The submitted manuscript is aimed to compare two different methods for oral tissue evaluation, in particular for periodontitis and periimplantitis.

The study is well organized and well conducted, and of wide interst for both dentists and pathologists.

However, the authors enrolled 22 patients only, so it is a pilot study (or a proof of concept ctudy). This should be clear for the readers, and possibly reported in the title.

Thank you, the title was changed.

No specific ethical issues are present in the reported study, but  please specify the University name and the date of the protocol approval at l. 98

The name of the University and date of the approval were integrated.

In sect. 2.2: please better specify the data regarding the manufacturers

The manufacturers’ details were added.

In sect. 3.1.1: comment more extensively the results and the tables' content.

A sentence was added in section 3.1.1.

In Discussion, the authors must report the limits of the study.

At the end of the discussion, a paragraph about the study limitations was added and implemented.

At last, a revision for English language is suggested.

Thank you, we had the English language revised.

Reviewer 2 Report

Excellent article, very clear, well presented, rigorous methodology !

Only one very minor comment: the legend of table 1 should be placed before the table and not within the text. 

Author Response

Excellent article, very clear, well presented, rigorous methodology !

Only one very minor comment: the legend of table 1 should be placed before the table and not within the text.

Thank you, we tried to format the legend in a more appropriate way.

Round 2

Reviewer 1 Report

The previous comments have been addressed and the paper is now acceptable